# Regional Association between Mean Air Temperature and Case Numbers of Multiple SARS-CoV-2 Lineages throughout the Pandemic

**DOI:** 10.3390/v14091913

**Published:** 2022-08-30

**Authors:** Camilla Mattiuzzi, Brandon M. Henry, Giuseppe Lippi

**Affiliations:** 1Service of Clinical Governance, Provincial Agency for Social and Sanitary Services (APSS), 38123 Trento, Italy; 2Clinical Laboratory, Division of Nephrology and Hypertension, Cincinnati Children’s Hospital Medical Center, Cincinnati, OH 45229, USA; 3Section of Clinical Biochemistry and School of Medicine, University of Verona, 37129 Verona, Italy

**Keywords:** SARS-CoV-2, COVID-19, temperature, infection

## Abstract

The association between mean air temperature and new SARS-CoV-2 case numbers throughout the ongoing coronavirus disease 2019 (COVID-19) pandemic was investigated to identify whether diverse SARS-CoV-2 lineages may exhibit diverse environmental behaviors. The number of new COVID-19 daily cases in the province of Verona was obtained from the Veneto Regional Healthcare Service, whilst the mean daily air temperature during the same period was retrieved from the Regional Agency for Ambient Prevention and Protection of Veneto. A significant inverse correlation was found between new COVID-19 daily cases and mean air temperature in Verona up to Omicron BA.1/BA.2 predominance (correlation coefficients between −0.79 and −0.41). The correlation then became positive when the Omicron BA.4/BA.5 lineages were prevalent (r = 0.32). When the median value (and interquartile range; IQR) of new COVID-19 daily cases recorded during the warmer period of the year in Verona (June–July) was compared across the three years of the pandemic, a gradual increase could be seen over time, from 1 (IQR, 0–2) in 2020, to 22 (IQR, 11–113) in 2021, up to 890 (IQR, 343–1345) in 2022. These results suggest that measures for preventing SARS-CoV-2 infection should not be completely abandoned during the warmer periods of the year.

## 1. Introduction

The severe acute respiratory syndrome coronavirus 2 (SARS-CoV-2), firstly identified at the end of 2019 in Wuhan (China), is the pathogen responsible for the coronavirus disease 2019 (COVID-19), one of the worst pandemics ever recognized throughout human history [1]. Several different aspects have been identified to support the high and rapid diffusion of this novel coronavirus all around the world compared to its predecessors that caused previous human coronavirus outbreaks such as SARS-CoV(-1) and Middle East respiratory syndrome coronavirus (MERS-CoV). These include some socioeconomic and environmental factors, such as the enormously magnified volume of migration, tourism, business trips [2], increased worldwide burden of air pollution [3], and increased population density [4], which all contribute to increase the likelihood of SARS-CoV-2 transportation and transmission. A number of biological issues may also contribute to potentiating inter-human diffusion of SARS-CoV-2 compared to other previously known coronaviruses. Compared to SARS-CoV(-1) and MERS-CoV, SARS-CoV-2 is characterized by an increased rate of asymptomatic and pre-symptomatic transmission [5], enhanced affinity for host-cell receptors, namely for the angiotensin converting enzyme 2 (ACE2) [6], and higher level of shedding in the upper respiratory tract, [7]. On the other hand, SARS-CoV-2 also expresses lower cumulative human pathogenicity compared to SARS-CoV and MERS-CoV, enhancing host survival and further spread of the virus during the infectious phase [8].

The relatively low resistance and stability of SARS-CoV-2 and other coronaviruses at high temperatures and/or heavy sunlight exposure suggests a potential for a seasonal pattern of infection. Specifically, human coronaviruses usually display the largest peak of infections during the cooler season, whilst the number of contagions typically decline (or even virtually disappear) during the warmest period of the year [9]. This basic assumption has been confirmed by a number of previous studies during the first two years of the ongoing COVID-19 pandemic, which have all underpinned the existence of an inverse correlation between air (outdoor) temperature and number of new COVID-19 diagnoses [10,11,12]. This strongly suggests that SARS-CoV-2 is indeed vulnerable to high temperatures and sustained ultraviolet (UV) irradiation [13,14]. Ulloa and colleagues showed that the combination of high temperature and strong UV irradiation was highly effective in rapidly (i.e., within minutes) reducing the number of SARS-CoV-2 viral particles belonging to the Alpha and Gamma lineages [15]. Nonetheless, the emergence of the new and highly mutated Omicron lineages (especially BA.1, BA.2, BA.2.12.1, BA.2.75, BA.4, and BA.5) at the end of the 2021 has almost challenged this previously straightforward concept, in that the burden of COVID-19 cases remained extraordinarily high in the Northern Hemisphere during the beginning of the 2022 summer [16], despite very hot temperatures and extreme heat waves in North America and Europe [17].

Hence, the purpose of this study was to investigate the association between mean air temperature and SARS-CoV-2 cases throughout the course of the COVID-19 pandemic, in order to identify whether the behavior of different SARS-CoV-2 lineages suggests a change in susceptibility to high temperatures.

## 2. Materials and Methods

The number of new COVID-19 daily cases diagnosed in the province of Verona (North-East Italy) throughout the SARS-CoV-2 pandemic (i.e., between 1 March 2020 and 31 July 2022) was obtained from online resources made available by the Healthcare Service of the Veneto Region [18], whilst the mean daily air temperature value recorded during the same period in Verona was retrieved from the official statistics available online from the website of Regional Agency for Ambient Prevention and Protection of Veneto (Agenzia Regionale per la Prevenzione e Protezione Ambientale del Veneto; ARPAV) [19]. The estimated prevalence of the different SARS-CoV-2 variants over time (i.e., >50% of all attributable cases) was defined using the official bulletin of the National Healthcare Institute (Istituto Superiore di Sanità; ISS) [20]. Results were presented as median and interquartile range (IQR). The association between mean air temperature and SARS-CoV-2 infections during the prevalence of the different SARS-CoV-2 variants was estimated using Spearman’s correlation. The statistical analysis was conducted with Analyse-it (Analyse-it Software Ltd., Leeds, UK).

The study was conducted in accordance with the Declaration of Helsinki, under the terms of relevant local legislation. This analysis was based on electronic searches in open and publicly available repositories, such that no informed consent or Ethical Committee approval was necessary.

## 3. Results

The prevalence of each SARS-CoV-2 variant over time (i.e., >50% estimated attributable cases) in the province of Verona is summarized in Table 1, along with the number of daily cases and the air temperature during the different periods.

The progressive number of new COVID-19 daily cases in the province of Verona along with the fluctuation of the mean air temperature in the same area throughout the study period is shown in Figure 1.

The Spearman’s correlation between the number of new daily COVID-19 cases sustained by different SARS-CoV-2 variants recorded in the province of Verona and the mean air temperature value in the same area and during the same period is shown in Table 2.

A statistically significant inverse correlation was found between the number of new COVID-19 daily cases and mean air temperature from the beginning of the pandemic until the period characterized by prevalence of the Omicron BA.1/BA.2 lineages, even though the values of the correlation coefficients tended to decrease over time. Although a negative association was still found between number of new daily COVID-19 cases and mean air temperature during predominance of the Omicron BA.1/BA.2 lineages in the area of Verona (r = −0.58), the correlation instead became positive when the Omicron BA.4/BA.5 lineages were seemingly prevalent (r = 0.32).

When the number of new COVID-19 daily cases diagnosed during the warmer period of the year in the province of Verona (i.e., typically between June and July) was compared across the three years of the pandemic, a gradual increase could be seen over time, from 1 (IQR, 0–2) new daily case in 2020, to 22 new daily cases (IQR, 11–113) in 2021, and up to 890 (IQR, 343–1345) new daily cases in 2022 (Figure 2), whilst the median value of the mean air temperature also increased from 22.2 (IQR, 20.0–24.6) °C in 2020, to 24.6 (IQR, 22.5–25.7) °C in 2021, and up to 27.0 (IQR, 25.8–28.0) °C in 2022.

Notably, the number of new COVID-19 daily cases in Verona was similar between the periods of Omicron BA.1/BA.2 and Omicron BA.4/BA.5 prevalence (i.e., 890 vs. 943), despite the mean air temperature value being nearly three-fold higher during the Omicron BA.4/BA.5 wave (i.e., 27.0 vs. 9.6 °C) (Table 1).

## 4. Discussion

Besides experiencing unusually high temperatures in the northern hemisphere, summer 2022 was also characterized by an unprecedented and almost unpredictable recrudescence of COVID-19 cases, most of which were seemingly attributable to the spread of the Omicron BA.4 and BA.5 lineages [21]. According to our analysis, for example, the median number of new COVID-19 daily cases diagnosed in the province of Verona increased from 1 in June–July 2020, to 22 in June–July 2021, up to 890 in June–July 2022, whilst the air temperature in the same area and comparable period of the year continuously rose throughout these three years (22.2 °C in 2020, 24.6 °C in 2021 and 27.0 °C in 2022, respectively).

This evidence, along with the disappearance of the previously well-known negative association between air (i.e., outdoor) temperature and SARS-CoV-2 infectivity, suggests that the many mutations characterizing the Omicron lineages BA.4/BA.5 may not only have contributed to increase the burden of vaccine breakthrough infections and re-infections [22], but may also have significantly modified their biological properties [23]. This was predictable, since the strong association between air temperature and SARS-CoV-2 infectivity that was seen during the outbreak sustained by the ancestral strain (r = −0.79) in Verona, tended to progressively decline over time, even before emergence of the Omicron BA.4/BA.5 lineages, thus testifying that the virus had already undergone a process of evolution, accumulating a vast array of specific mutations that have rendered it more infective, less sensitive to humoral immunity and neutralization [24], and thus perhaps even more resistant to theoretically “unfavorable” environmental conditions, namely heat and UV light exposure [23]. Alternatively, the extremely high transmissibility of the BA.4 and BA.5 lineages may marginalize any impact of the environment on its infectivity.

It hence noteworthy that lower vulnerability to high temperature or sunlight exposure has already been advocated as one of the potential mechanisms explaining—at least partially—the enhanced infectivity of the early Omicron lineages. More specifically, Hirose et al. assessed the environmental stability over time of several different SARS-CoV-2 variants on plastic and human skin surfaces [25], reporting that the median survival time on human skin surface was 8.6 h for the ancestral (i.e., the prototype Wuhan) lineage, increasing to 19.6 h for the Alpha variant, and up to 21.1 and 22.5 h for the Delta and Omicron BA.1/BA.2 lineages. Similar evidence has been provided by Chin et al. [26], who also reported that the Omicron BA.1 strain displayed higher stability on smooth and porous surfaces compared to the ancestral SARS-CoV-2 variant. Unfortunately, to the best of our knowledge, no study has thus far explored the resistance (or vulnerability) of Omicron BA.4 and/or BA.5 under different environmental conditions. We can therefore only epidemiologically infer that these two new lineages may exhibit an even improved stability at warm temperatures.

## 5. Conclusions

Although we acknowledge that publicly available new COVID-19 case numbers may not straightforwardly reflect the true prevalence in the population, especially in the later pandemic phase when more patients self-test and do not report to government authorities and some preventive measured have been gradually abandoned, the results of our analysis suggest that SARS-CoV-2 has increased its capacity to spread at different temperatures over time, even during weather conditions generally considered unfavorable (i.e., warmer temperatures). These findings may have important public-health implications. First, consideration should be given to not completely abandoning some well-known and widely adopted measures for preventing SARS-CoV-2 infection (e.g., use of face masks, hand hygiene, and avoiding large mass gatherings) during the warmer periods of the year, as has become rather evident in a large number of countries [27]. Additionally, healthcare systems must be prepared to face a potential resurgence of COVID-19-related hospitalizations even during the warmer periods, in order to avoid the perfect storm that could be caused by concomitantly increased healthcare pressure due to hot temperatures [28] and SARS-CoV-2 severe/critical infections [29].

## Figures and Tables

**Figure 1 viruses-14-01913-f001:**
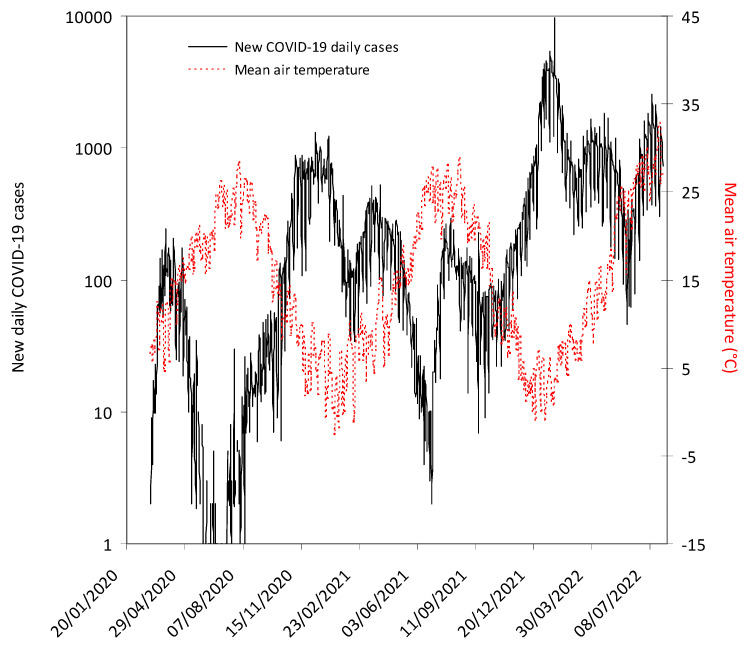
The number of new coronavirus disease 2019 (COVID-19) daily cases recorded in the province of Verona and fluctuation of mean air temperature in the same area throughout the study period. The grey area indicates the lockdown period in Italy (9 March 2020–18 May 2020). The mandatory requirement to wear face mask in open spaces and within closed areas (except healthcare facilities) was definitively abrogated from 11 February 2022 and 15 June 2022, respectively.

**Figure 2 viruses-14-01913-f002:**
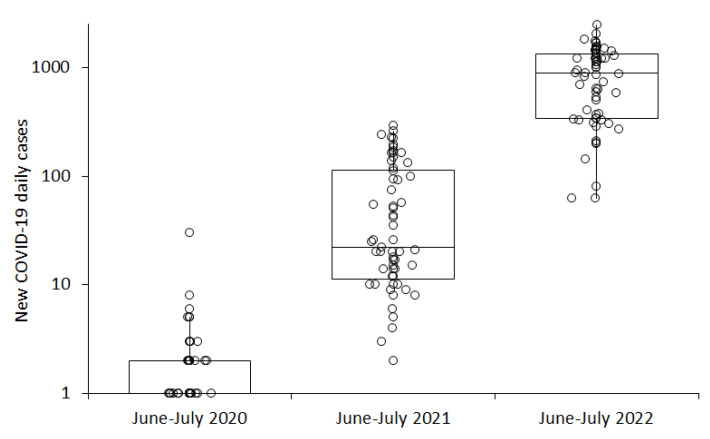
Number of new coronavirus disease 2019 (COVID-19) daily cases recorded in the province of Verona in June–July of the years 2020, 2021, and 2022. Results are presented as median and interquartile range (IQR).

**Table 1 viruses-14-01913-t001:** Period of prevalence of different severe acute respiratory syndrome coronavirus 2 (SARS-CoV-2) variants in the province of Verona along with number of new daily COVID-19 cases and air temperature in the same area. Data are shown as median and interquartile range (IQR).

SARS-CoV-2 Variant	Period	New COVID-19 Daily Cases (IQR)	Mean Air Temperature (IQR)
Ancestral	March 2020–January 2021	44 (7–208)	14.6 (7.5–20.4) °C
Alpha	February 2021–June 2021	111 (37–228)	12.4 (7.9–17.5) °C
Delta	July 2021–December 2021	114 (70–191)	17.4 (8.4–22.7) °C
Omicron BA.1/BA.2	January 2022–May 2022	943 (572–1385)	9.6 (5.9–14.8) °C
Omicron BA.4/BA.5	June 2022–July 2022	890 (343–1345)	27.0 (25.8–28.0) °C

**Table 2 viruses-14-01913-t002:** Spearman’s correlation between the number of new daily coronavirus disease 2019 (COVID-19) cases recorded in the province of Verona and the mean air temperature in the same area according to the period of prevalence of the different severe acute respiratory syndrome coronavirus 2 (SARS-CoV-2) variants.

SARS-CoV-2 Variant	Spearman’s Correlation (95% CI)
Ancestral	−0.79 (−0.83 to −0.75; *p* < 0.001)
Alpha	−0.63 (−0.72 to −0.52; *p* < 0.001)
Delta	−0.41 (−0.52 to −0.28; *p* < 0.001)
Omicron BA.1/BA.2	−0.58 (−0.68 to −0.47; *p* < 0.001)
Omicron BA.4/BA.5	0.32 (0.08 to 0.53; *p* = 0.011)

## Data Availability

Data are available upon reasonable request to the corresponding author.

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
