# Peer review of "Regional Association between Mean Air Temperature and Case Numbers of Multiple SARS-CoV-2 Lineages throughout the Pandemic"

_viruses, 2022, doi:10.3390/v14091913_

Round 1
Reviewer 1 Report
Mattiuzzi et al., in this research article investigate the association between mean air temperature and SARS-CoV-2 infectivity throughout the ongoing coronavirus disease 2019 (COVID-19) pandemic in order to identify whether diverse SARS-CoV-2 lineages may exhibit different environmental behaviors. The study appears to be interesting despite the fact that it doesn't seem to add new details regarding our current knowledge. In fact the results obtained reinforced that SARS-CoV-2 has increased its capacity to spread at different temperatures over time, even during weather conditions generally considered unfavorable (i.e., warm climate) and to demonstrate this the authors focus on the epidemiological assessment of climatic and case count data in the region of Verona, North Italy. Although those findings may have important public health implications. I would be curious to see which one is supposed to be the behaviour of those data in other Italian regions, in order to identify possible paths of spreading.
Author Response
Mattiuzzi et al., in this research article investigate the association between mean air temperature and SARS-CoV-2 infectivity throughout the ongoing coronavirus disease 2019 (COVID-19) pandemic in order to identify whether diverse SARS-CoV-2 lineages may exhibit different environmental behaviors. The study appears to be interesting despite the fact that it doesn't seem to add new details regarding our current knowledge. In fact the results obtained reinforced that SARS-CoV-2 has increased its capacity to spread at different temperatures over time, even during weather conditions generally considered unfavorable (i.e., warm climate) and to demonstrate this the authors focus on the epidemiological assessment of climatic and case count data in the region of Verona, North Italy.
- We are thankful to the referee for the globally favourable comments on our manuscript. Although we basically agree with the referee that “The study appears to be interesting despite the fact that it doesn't seem to add new details regarding our current knowledge”, we would like to highlight here that this is the first report (to the best of our knowledge) reporting this information based on OFFICIAL data (i.e., data captured from the Healthcare Service of the Veneto Region and the Regional Agency for Ambient Prevention and Protection of Veneto). This determines high reliability in our statistics.
Although those findings may have important public health implications. I would be curious to see which one is supposed to be the behaviour of those data in other Italian regions, in order to identify possible paths of spreading.
- ANSWER: This is a very good point but, unfortunately, we have no access to the official metrological data of other Regional Agency for Ambient Prevention and Protection except those for the Veneto Region. Moreover, the other regional agencies use different criteria for classifying COVID-19 positive cases compared to those use by the Healthcare Service of the Veneto Region, thus making the direct comparison totally unreliable. We agree indeed with the valuable comment of the referee, but we prefer to limit our analysis to our geographical location to avoid providing unreliable statistics that would then bias the entire work. It shall also be considered that extending the area of analysis would complicate the interpretation of the findings, since the temperature in Italy varies considerably from one region to another, sometimes by over 10°C. We hope that the referee would finally understand out position of not adding potentially misleading data to our analysis.
Reviewer 2 Report
The study analyzed the association between mean air temperature and SARS-CoV-2 case numbers during the COVID-19 pandemic in the province of Verona of Italy. Results showed inverse correlation between new cases and mean air temperature up to Omicron BA.1/BA.2 predominance, while the correlation then became positive when the Omicron BA.4/BA.5 lineages were prevalent.
Although the findings are of interest and value to readers and the medical community, there are issues that need to be addressed.
1. A title that better reflects the study findings would read “Regional association between mean air temperature and case number of multiple SARS- 2 CoV-2 lineages throughout the pandemic”.
2. Abstract, line 12. Similar to the title, only case numbers are reported in this article rather than infectivity, so the word infectivity can be replaced by new case numbers.
3. Figure 1. In addition to simply the time on X-axis, lockdown policies vary throughout the pandemic, so it would be more helpful to indicate major lockdown and social distancing policies along with the timeline.
4. Publicly available new case numbers may not truly represent prevalence in the population, particularly in later pandemic phase when more patients self-test and do not report to government authorities. This caveat needs to be discussed.
Author Response
The study analyzed the association between mean air temperature and SARS-CoV-2 case numbers during the COVID-19 pandemic in the province of Verona of Italy. Results showed inverse correlation between new cases and mean air temperature up to Omicron BA.1/BA.2 predominance, while the correlation then became positive when the Omicron BA.4/BA.5 lineages were prevalent. Although the findings are of interest and value to readers and the medical community, there are issues that need to be addressed.
- We are thankful to the referee for the globally favourable comments on our manuscript. We’ll do our best to improve it according to the referee’s suggestions.
- A title that better reflects the study findings would read “Regional association between mean air temperature and case number of multiple SARS- 2 CoV-2 lineages throughout the pandemic”.
ANSWER: We definitely agree and the title has been thoughtfully revised as suggested: “Regional association between mean air temperature and case number of multiple SARS- 2 CoV-2 lineages throughout the pandemic”.
- Abstract, line 12. Similar to the title, only case numbers are reported in this article rather than infectivity, so the word infectivity can be replaced by new case numbers.
ANSWER: We definitely agree and the title has been thoughtfully revised as suggested.
- Figure 1. In addition to simply the time on X-axis, lockdown policies vary throughout the pandemic, so it would be more helpful to indicate major lockdown and social distancing policies along with the timeline.
ANSWER: Good point, thanks. Figure 1 revised accordingly, as suggested. The Following explanation has also been included in the caption of the figure: “The grey area indicates the lockdown period in Italy (March 9, 2020 – May 18, 2020). The mandatory requirement to wear face mask in open spaces and within close areas (except healthcare facilities) has been definitively abrogated from February 11, 2022 and June 15, 2022, respectively”.
- Publicly available new case numbers may not truly represent prevalence in the population, particularly in later pandemic phase when more patients self-test and do not report to government authorities. This caveat needs to be discussed.
ANSWER: This is truly a very good point. We fully agree and we have in fact included this aspect as a limitation in our study, as follows: “Although we acknowledged that publicly available new COVID-19 case numbers may not straightforwardly reflect the true prevalence in the population, especially in later pandemic phase when more patients self-test and do not report to government authorities, …”.
Round 2
Reviewer 1 Report
Authors have raised the point enhanced during the first round of revision. The present manuscript can be accepted for publication